Molecular modeling and pharmacophore elucidation study of the Classical Swine Fever virus helicase as a promising pharmacological target

Vlachakis Dimitrios
Kossida Sophia skossida@bioacademy.gr
Bioinformatics & Medical Informatics Team, Biomedical Research Foundation , Academy of Athens, Athens , Greece
Peng Yonghong
Electronic publication date: 2013 Jun 11
Publication date: 2013
Volume: 1
Electronic Location ID: e85
Received 2013 Apr 4; Accepted 2013 May 21
Copyright: © 2013 Vlachakis & Kossida
Copyright year: 2013
Copyright holder: Vlachakis & Kossida
License: This is an open access article distributed under the terms of the Creative Commons Attribution License, which permits unrestricted use, distribution, and reproduction in any medium, provided the original author and source are credited.
License URL: https://creativecommons.org/licenses/by/3.0/

Keywords: Classical Swine Fever virus, Molecular modeling, Bioinformatics, Pharmacophore, Antiviral

Funding: The authors received no funding for this study.

==============================
The Classical Swine Fever virus (CSFV) is a major pathogen of livestock and belongs to the flaviviridae viral family. Even though there aren’t any verified zoonosis cases yet, the outcomes of CSFV epidemics have been devastating to local communities. In an effort to shed light to the molecular mechanisms underlying the structural and drug design potential of the viral helicase, the three dimensional structure of CSFV helicase has been modeled using conventional homology modeling techniques and the crystal structure of the Hepatitis C virus (HCV) as a template. The established structure of the CSFV helicase has been in silico evaluated for its viability using a repertoire of in silico tools. The ultimate goal of this study is to introduce the 3D conformation of the CSFV helicase as a reliable structure that may be used as the designing platform for de novo, structure-based drug design experiments. In this direction using the modeled structure of CSVF helicase, a 3D pharmacophore was designed. The pharmacophore comprises of a series of key characteristics that molecular inhibitors must satisfy in order to achieve maximum predicted affinity for the given enzyme. Overall, invaluable insights and conclusions are drawn from this structural study of the CSFV helicase, which may provide the scientific community with the founding plinth in the fight against CSFV infections through the perspective of the CSFV helicase as a potential pharmacological target. Notably, to date no antiviral agent is available against the CSFV nor is expected soon. Subsequently, there is urgent need for new modern and state-of-the-art antiviral strategies to be developed.

Introduction

The viral family Flaviviridae comprises the genera Flavivirus, Pestivirus and Hepacivirus and includes numerous important human and animal pathogens (Nulf & Corey, 2004). The small, enveloped virions of the different members of the Flaviviridae family contain a single-stranded, positive-sense RNA genome of about 9.5–12.5 kb. The genome consists of a single, long open reading frame (ORF), which is flanked by untranslated regions (UTRs) at the 5′ and 3′ ends. Recent studies on sub-genomic Pestivirus and Flavivirus RNA replicons have revealed that the non-structural (NS) proteins, which are encoded by the C-terminal part of the polyprotein, play a crucial role in viral RNA replication (Nulf & Corey, 2004). Accordingly, these proteins are assumed to form replication complexes in conjunction with genomic RNA and possibly with other cellular factors.

Classical Swine Fever virus (CSFV) is a member of the Flaviviridae virus family that affects livestock and therefore represents a very important pathogen (Calisher & Gould, 2003). Even though CSFV may not be a zoonotic disease that is capable of infecting humans, the consequences of its epidemics are devastating, since livestock is closely related with the economic and social well being of many countries in the world (Behrens et al., 1998). To date no antiviral agent is available nor is expected soon. There are quite a few vaccines currently available in the market for some strains of CSFV, but reportedly the virus is becoming quite persistent and even under extensive vaccination there are many cases of reoccurrence of the infection upon vaccination (Wu et al., 2005). Consequently, there is urgent need for new antiviral strategies to be developed. In this direction, the non-structural NS3 viral helicase constitutes an ideal target for structural studies towards the establishment of a comprehensive in silico platform for de novo inhibitor design. The viral helicase is responsible for the unwinding of the viral genetic material (Phoon et al., 2001). Thus the viral NS3 helicase is a key enzyme for the survival, propagation, proliferation and finally transmission of the virus. Sequence alignments of the Classical Swine Fever viral helicase identified several conserved sequence motifs that are important for biological functions. So far, the crystal structures of helicases from various RNA viruses have been determined, including the helicases from the Hepatitis C, Dengue, Yellow Fever, and Kunjin viruses (Diana & Bailey, 1997).

Herein, the 3D model of the CSFV helicase structure has been established using conventional homology modeling techniques and the X-ray determined 3D-structure of the Hepatitis C virus helicase as a template. In order to structurally improve the quality of the homology model, it was subjected to exhaustive molecular dynamics simulations (MDS). The reliability and viability of the helicase model as a plausible platform for structure based drug design experiments was confirmed using a repertoire of in silico scoring tools, including MOE, Procheck and Verify3D. It was determined that the established 3D model of the helicase enzyme of CSFV exhibits all known structural motifs that are unique to the helicase enzymes of the flaviviridae virus family and therefore could provide the platform for further drug design experiments. Towards this direction an in silico structure-based drug design experiment was conducted, which led to the establishment of a 3D pharmacophore model that is highly specific for the helicase enzyme of CSFV. The pharmacophore model may be used in future experiments involving the high virtual throughput screening of large molecular databases towards the identification of novel anti-CSFV agents.

Methods

Coordinate preparation and model availability

3D coordinates were obtained from the X-ray solved, crystal structures of HCV helicases with RCSB codes: 1A1V and 8OHM. The 8OHM entry is the full length, unbound form of HCV helicase, whereas the 1A1V entry contains the full length HCV helicase co-crystalized with a single strand of an oligonucleotide. All generated 3D CSFV helicase models in this study are available via email request to the corresponding author. All generated 3D homology models are available to download at: http://www.bioacademy.gr/bioinformatics/csfv/index.html

Sequence alignment

The amino acid sequence of Classical Swine Fever viral helicase was obtained from the GenBank database (accession no: NC_006551, entry name: Classical Swine Fever virus, complete genome). Using the Gapped-BLAST (Altschul et al., 1997) through NCBI (Benson et al., 2007) the homologous species of the Hepatitis C virus helicase was identified, which was used as a template for the homology modeling of the Classical Swine Fever viral helicase. The structure of the HCV helicase has been determined via X-ray crystallography (Kim et al., 1998). The sequence alignment was done using the online version of ClustalW (Thompson, Higgins & Gibson, 1994). The alignment was repeated using Hidden Markov Models and the result was the same as the one obtained by ClustalW, due to the fact that there are several anchoring conserved motifs throughout the alignment (Eddy, 1995).

Homology modeling

The homology modeling of the Classical Swine Fever viral helicase was carried out using the Modeller package (version 9.10) (Sali et al., 1995). The RCSB entries 1A1V and 8OHM were both used as template structures. A different model was generated for each one of the two available HCV crystal structure. The sequence alignment between the raw sequence of the Classical Swine Fever and the full sequence of the 1A1V HCV helicase revealed almost 32% identity, whereas the identity percentage between the Classical Swine Fever and the full sequence of 8OHM HCV helicase does not exceed 35%. The sequence alignment identity percentage between the two HCV template structures is 93.33% due to the fact that these sequences originate from different HCV strains. The above mentioned identity scores are on the lower boundary at which conventional homology modeling techniques are applicable. The model that was based on the 1A1V HCV structure was chosen to be used for the purposes of this study, due to the fact that this structure file has been co-crystallized with a single-stranded oligonucleotide. The homology model method of Modeller comprises the following steps: First, an initial partial geometry specification, where an initial partial geometry for each target sequence is copied from regions of one or more template chains. Second, the insertions and deletions task, where residues that still have no assigned backbone coordinates are modelled. Those residues may be in loops (insertions in the model with respect to the template), they may be outgaps (residues in a model sequence which are aligned before the C-terminus or after the N-terminus of its template) or may be deletions (regions where the template has an insertion with respect to the model). For this study, though, outgaps have not been included in the homology modeling process. The third step is the loop selection and sidechain packing where a collection of independent models is created. The last step is the final model selection and refinement, where the final models are scored and ranked, after they have been stereochemically checked with the “Protein Geometry” module for persisting errors. Finally, necessary secondary structure predictions were performed using the NPS (Network Protein Sequence Analysis) web-server and the GeneSilico MetaServer, which confirmed the choice of the Hepatitis C virus helicase as a template for this study (http://npsa-pbil.ibcp.fr).

Molecular electrostatic potential (MEP)

Electrostatic potential surfaces were calculated by solving the nonlinear Poisson-Boltzmann equation using the finite difference method as implemented in the PyMOL Software (DeLano, 2002). The potential was calculated on grid points per side (65, 65, 65) and the grid fill by solute parameter was set to 80%. The dielectric constants of the solvent and the solute were set to 80.0 and 2.0, respectively. An ionic exclusion radius of 2.0 Å, a solvent radius of 1.4 Å and a solvent ionic strength of 0.145 M were applied. Amber99 (Duan et al., 2003) charges and atomic radii were used for this calculation.

Energy minimization and molecular dynamics simulations

Energy minimizations were used to remove any residual geometrical strain in each molecular system, using the Charmm27 forcefield as it is implemented into the Gromacs suite, version 4.5.5 (Hess et al., 2008). All Gromacs-related simulations were performed though our previously developed graphical interface (Sellis, Vlachakis & Vlassi, 2009). Molecular systems were then subjected to unrestrained MDS using the Gromacs suite, version 4.5.5 (Hess et al., 2008). MDS took place in an SPC water-solvated, periodic environment. Water molecules were added using the truncated octahedron box extending 7 Å from each atom. Molecular systems were neutralized with counter-ions as required. For the purposes of this study all MDS were performed using the NVT ensemble in a canonical environment, at 300 K and a step size equal to 2 fs for a total 100 ns simulation time. An NVT ensemble requires that the Number of atoms, Volume and Temperature remain constant throughout the simulation.

Model evaluation

Evaluation of the model quality and reliability in terms of its 3D structural conformation is very crucial for the viability of this study. Therefore, the produced models were initially evaluated within the Gromacs package by a residue packing quality function which depends on the number of buried non-polar side chain groups and on hydrogen bonding. Moreover, the suite PROCHECK (Laskowski et al., 1996) was employed to further evaluate the quality of the produced Classical Swine Fever virus helicase model. Verify3D (Eisenberg, Luthy & Bowie, 1997) was also used to evaluate whether the model of Classical Swine Fever virus helicase is similar to known protein structures. Finally, the Molecular Operating Environment (MOE) suite was used to evaluate the 3D geometry of the models in terms of their Ramachandran plots, omega torsion profiles, phi/psi angles, planarity, C-beta torsion angles and rotamer strain energy profiles.

Structure-based drug design

The design of the new series of molecules via structure-based drug design was conducted using Ligbuilder (version 1.2). This program will start from a “seed” compound that must be manually positioned in the 3D conformational space of the molecular system (Fig. 6). For the purposes of this study, the mercaptoethanol co-crystallized compound that was attached to the Cys431 residue of the HCV structure was used as seed structure. The structures of CSFV and HCV helicase were structurally superimposed and their coordinates recalculated within the same Cartesian molecular system. First the S-S bond between the attached compound on the Cys431 and the Cys431 was broken and the hydrogens were restored from where they were missing. Then the oxygen was removed since its existence would significantly reduce the number of fragments suitable for that seed-receptor arrangement. The remaining compound was used as a starting point for the “growing” algorithm of LigBuilder. The complex was energetically minimized using a molecular mechanics algorithm, having fixed the backbone of the protein. The detached compound was entered to be the starting point of the drug design algorithm and thus it was expected that this moiety would be present in this position on all the new compounds, as it was already known from the HCV crystal structure that it was capable of interacting with the Cys431 amino acid. The algorithm used the mercaptoethanol compound as a starting point and started to grow structures by combining different chemical fragments that it stores in its database. The criteria are to optimally utilize the available space of the receptor and to establish the maximum amount of interactions with the adjacent residues of the helicase. All the different compounds that were designed were deposited in a folder for further investigation. A similarity cut-off of 90% was used in order to make sure that structurally all the different compounds in that folder would be at least 90% different. The space available was filled-in with newly designed compounds, with the only size-limiting parameter being the pre-defined molecular weight of the compound. After 30 generations of the Ligbuilder growing genetic algorithm, a set of 1200 compounds was obtained, custom designed for the ssRNA channel of the CSFV helicase.

Docking

In order to establish the complex structures of the CSVF helicase in silico and each one of the 1200 potential inhibitor compounds, the docking suite ZDOCK (version 3.0) was used (Chen, Li & Weng, 2003). Docking experiments were conducted on the models that had been energetically minimized and conformationally optimized using molecular dynamics simulations. ZDOCK is a protein-protein docking suite that utilizes a grid-based representation of the molecular system involved. In order to efficiently explore the search space and docking positions of the molecules as rigid bodies, ZDOCK takes full advantage of a three-dimensional fast Fourier transformation algorithm. It uses a scoring function that returns electrostatic, hydrophobic and desolvation energies as well as performing a fast pairwise shape complementarity evaluation. Moreover it uses the contact propensities of transient complexes of proteins to perform an evaluation of a pairwise atomic statistical potential for the docking molecular system. RDOCK was utilized to refine and quickly evaluate the results obtained by ZDOCK (Li, Chen & Weng, 2003). RDOCK performs a fast minimization step to the ZDOCK molecular complex outputs and re-ranks them according to their re-calculated binding free energies.

Hybrid QSAR analysis & pharmacophore elucidation

In this study a hybrid QSAR study was conducted using the 1200 compounds from the previous drug design experiment. Since no biological activity data was available for these molecules, an in silico predicted Ki score was used to rank them that was primarily based on their docking scores. The aim of this analysis was initially to interpret the unique characteristics of various compounds in regards to their estimated activities and to provide the means required to establish a 3D-pharmacophore model that would enable us to more accurately screen for anti CSFV agents.

We used the set of the 1200 structurally distinct inhibitor compounds, which was generated by the structure based drug design algorithm. The predicted inhibitory potential of those compounds, was associated with their conformations as they were submitted to the Pharmacophore Elucidation Query module of MOE (Group CC, 2012). The algorithm initially identifies all features common to the highest ranking compounds, as they were scored by the Ligbuilder scoring algorithm, as well as features present in the least ranking compounds. The first ones are retained, whereas the latter are discarded. Finally, a set of regression parameters are used to estimate the activity value of each compound in the training set. The relationship between the geometric fit value and activity value is utilized for this computation. Pharmacophore hypotheses showing best correlation in the 3D arrangement of features in a given training set compounds with the corresponding pharmacological activities are formed and ranked. Several structure activity relationship (SAR) pharmacophore models were derived from the training set of compounds.

Results & discussion

The NS3 domain of Flaviviridae contains both the protease and the helicase coding regions. For the purposes of this study, only the helicase protein of the Classical Swine Fever virus was used and aligned with the Hepatitis C virus helicase structure that were selected to be used as templates. Notably, all major helicase motifs, which are characteristic and unique to the helicases of the flaviviridae viral family were found to be completely conserved (Fig. 1).

Figure 1 Primary sequence alignment of CSVF against the HCV template (RCSB entry: 1A1V).

All seven major conserved motifs of Flaviviridae helicases have been highlighted in red. The key conserved amino acids that will be used as targets for the drug design steps have been highlighted in bold and magenta.

The 1A1V HCV helicase structure has been established by X-ray crystallography at 2.20 Å resolution, while the 8OHM at 2.30 Å (Kim et al., 1998; Cho et al., 1998). The HCV helicase was selected as the most suitable template for the homology modeling of the Classical Swine Fever virus. Both CSFV and HCV belong to the same viral family and upon a blastp search on the Protein Databank it was found that 1A1V and 8OHM bear the highest sequence percentage identity to CSFV (Fig. 1). Furthermore, the secondary structure prediction for the Classical Swine Fever virus helicase was found to be very analogous to the secondary structure of the HCV helicase (data not shown). More specifically, protein fold recognition techniques aim to identify and pinpoint similarities among 3D protein structures that are not supplemented by significant sequence similarity. The suitability of HCV as the best available template was also confirmed by a fold recognition (FR) methodologies. The underlying principle behind FR techniques is that a quick search for protein folds is made in large protein databases, which is looking to identify folds that are compatible with a particular sequence. Unlike simple comparisons based on sequence only, these more sophisticated methods exploit all the extra 3D structural information that is readily available for many proteins. In essence these techniques turn the protein folding problem around: rather than predicting how a sequence will fold, they predict how well a fold will fit a sequence (Rost, Schneider & Sander, 1997). The CSVF homology modeling study constitutes one of these striking examples of structurally and functionally identical enzymes which only share a low primary sequence identity. In order to confirm that the HCV helicase is indeed the best template choice we used an in-house developed platform that performs protein similarity searches based on secondary structural information rather than primary sequence searches.

The first structural superimposition between the CSFV model and its HCV template exhibited an alpha-carbon RMSD that fall well within 0.64 angstroms (Fig. 2). The CSFV model was consequently checked with PROCHECK for its geometry mathematical accuracy. In addition to that, the Verify3D algorithm was employed for a more in-depth evaluation for its structure. A direct compatibility comparison between the 3D model of Classical Swine Fever virus helicase to its own amino acid sequence was performed by Verify3D. Judging strictly on location and environment, each residue is assigned a structural class. In order to do this a rather large database of reference structures is being used as a control. The Classical Swine Fever virus helicase model scored a very reliable range between + 0.33 and + 0.68. This was further confirmation in that the established CSVF 3D model is of high quality and mathematically reliable. Verify3D scores that fall below the +0.1 mark are indicative of major problems in the structure of the model as it can be mathematically evaluated (Eisenberg, Luthy & Bowie, 1997). The quality of the CSFV model was finally empirically confirmed in terms of its structural compliance to its HCV template and all known unique characteristics of the helicases of the Flaviviridae viral family. More precisely the helicase enzymes of the flaviviridae virus family belong to the helicase superfamily II which bear seven common motifs within their domains. All those motifs have been structurally conserved on the Classical Swine Fever virus helicase model. They share the same 3D spatial coordinates to their corresponding motifs located on the X-ray crystal helicase structures of the HCV, Kunjun and Dengue viruses (Kim et al., 1998; Khromykh & Westaway, 1997; Xu et al., 2005). More importantly the ssRNA interacting residues have been structurally conserved on the CSFV model (Fig. S1).

Figure 2 The CSFV helicase homology model.

(A) Ribbon representation of the produced Classical Swine Fever virus helicase model. (B) The Classical Swine Fever virus helicase model superimposed with its Hepatitis C virus helicase template (RCSB entry: 1A1V). The Classical Swine Fever virus helicase model is in red, whereas the Hepatitis C virus helicase template is in green and the ssRNA substrate is in blue.

Description of the Classical Swine Fever virus helicase model

As anticipated from the initial sequence alignment (Fig. 1) and the consecutive secondary structure prediction analysis (data not shown), the Classical Swine Fever virus helicase model has retained all structural features that characterize flaviviridae helicases. The GxGKT/S motif of helicase domain 1 is a Walker A motif that is also conserved in kinases. Its role is to aid with the binding of the β-phosphate of ATP [REF]. Evidence from mutagenesis studies confirms that any changes within this motif will yield an inactive helicase. The conserved DExH motif, which is involved in the Mg2+ ion and ATP interaction and hydrolysis, is equally important. Findings from both adenylate and thymidine kinases confirm that an aspartic acid residue is responsible for the Mg2 + ion coordination that lead to a conformational layout that optimizes the ATP molecule’s orientation for a highly efficient nucleophilic attack (Kim et al., 1998). On the other hand the QRxGRxGR motif is not associated with ATP hydrolysis rather than the establishment of direct specific interactions between the helicase cleft and the nucleic acid during the unwinding process (Kim et al., 1998). Overall it was found that the CSVF helicase model has shape, size and topology identical to that of its template (Figs. 2 and 3). The domain layout arrangement of the model is in agreement with the general flaviviridae helicase pattern. First and third domains are more closely associated and are separated by a rather large channel from domain two. This channel is the ssRNA cleft, where the processing of the oligonucleotide takes places and unwinding occurs. During the latter process, domain two undergoes substantial conformational changes. Moreover it moves away from domains one and three as the strength of specific interaction in the ssRNA channel must be weakened for the translocation process to occur. Figure 3 depicts the conservation of the ATP and the ssRNA channels between the CSVF model and its template HCV X-ray determined helicase structure (Kim et al., 1998).

Figure 3 Structural location of the conserved motifs.

The Hepatitis C virus helicase (in green color) is structurally superposed over the Classical Swine Fever virus helicase model (colored in Red). The major motifs have been color-coded according to the conventions of Fig. 1, and are showing in space filling representation. The ssRNA oligonucleotide is showing in blue ribbon representation. The white arrow point to the ATP site of the helicase. Notably, the conserved motifs sit around the ATP site, where they coordinate the ATP hydrolysis by the helicase enzyme.

ssRNA – ATP substrates and MD simulations

In an effort to confirm the functionality, suitability and reliability of the CSFV helicase model to be used in structure-based drug design experiments, the specific interactions with the ssRNA substrate were analyzed (Fig. S1). The coordinates of the ssRNA molecule as well as those for the ADP molecule and the Mg2 + ions were copied from the HCV template structure. Then exhaustive molecular dynamics simulations were performed to the CSVF model in the presence of all substrate molecules, in an explicitly solvated periodic box with SPC water molecules. Post molecular dynamics analysis confirmed that the majority of the protein and the ssRNA substrate interactions are established between the backbone of the oligonucleotide and a series of conserved residues on the helicase channel. Energy equilibrium is attained almost simultaneously for both the model and the template helicases. The alpha carbon root mean square deviation (CαRMSd) between equivalent atoms of the HCV and the post-MDs CSFV helicases does not exceed 0.37 Å. While large RMSd values are indicative of systems of poor quality, the low RMSd values between the CSVF model and the HCV template indicate that their 3D structures remain conformationally similar, thus reflecting the high quality of the post molecular dynamics CSVF helicase model. Moreover, the Ramachandran plots of the HCV template and the CSFV model confirm that there are no residues in the disallowed regions of the plot (Fig. 4). The Ramachandran plot is the most powerful and most established in silico tool for the stereochemical evaluation of the quality of a protein’s backbone, judging from its phi/psi dihedral angles. The phi/psi torsion angles cluster in favorable regions of the plot for the Classical Swine Fever virus helicase 3D model. None of those torsions duel in disallowed regions of the plot, which correspond to residues having steric hindrance and bad backbone geometries. The Classical Swine Fever virus helicase model residues that are found in the allowed regions of the plot are rendered in green whereas the generously allowed regions contain residues colored yellow. No residues are located in disallowed regions of the Ramachandran plot, a fact that verifies the quality of the established Classical Swine Fever virus helicase 3D model. Likewise, the dihedral plots of the omega, planarity, C-beta angles and energy packing per residue are all within the allowed regions, below the disallowed threshold (Figs. S2 and S3). Taken together, all of the above confirm the viability and suitability of the homology modeling of the Classical Swine Fever virus helicase model to be used in further structure based in silico drug design experiments.

Figure 4 The Ramachandran plots for the HCV template structure (PDB entry: 1A1V) on the top and the CSFV model after the molecular dynamics simulations below.

Electrostatic potential surfaces

The molecular surface of the produced Classical Swine Fever virus helicase model was analyzed by calculating its electrostatic potential (Fig. 5). The aim of this study was to enable us to perform direct structural comparisons between the CSFV model and the HCV template structures it came from. Consequently, the electrostatic potential surface was calculated also for both template HCV helicase structures that were used (Fig. 5). The CSVF model and the two helicases exhibited almost identical electrostatic surfaces. More precisely, the model and its templates share all key common structural features vital to viral helicases. They have a negatively charged ssRNA entrance to the helicase tunnel and a clearly identical ATP hydrolysis site. Taken together these similarities confirm the validity and reliability of the model which can be used for structure based drug design studies with high levels of confidence.

Figure 5 Molecular electrostatic potential surfaces.

Electrostatic surfaces potential for the Classical Swine Fever virus helicase model and the Hepatitis C virus helicase template (X-ray structure: 1A1V).

Structure based de novo drug design

The area surrounding the Cys431 residue was selected to be used as the target site for the drug design experiment for the purposes of this study. One of the HCV crystal helicase structures has been co-crystallized with a single molecule of mercaptoethanol via a disulphide bond. The Cys431 residue has established a S-S bond with a mercaptoethanol (S-CH4-CH4-OH) in the HCV helicase X-ray structure. That can only mean that the Cys431 is accessible to the solvent, even when ssRNA is present, and could potentially be targeted for the establishment of interactions with a future inhibitor. Notably, Cys431 is located in a very strategic position to block the passage of the ssRNA through the helicase (Fig. 6). Prior study of the HCV area surrounding the Cys431 residue has indeed confirmed the ability and potential of this residue to covalently bond with an HCV inhibitor and to interact with the two flanking Arginine residues (Arg393 and Arg481). The latter HCV inhibitor exerted activity in the micromolar range (Kandil et al., 2009; Vlachakis, Koumandou & Kossida, 2013). These residues were set to define the active site of the helicase that was going to be targeted later on. The arginine residues were expected to establish H-bonds, whereas the Cys431 residue was expected to establish an S-S or an S-C bond with the inhibitor (Fig. 7). Accordingly, upon structural superimposition of the CSVF helicase model and the HCV crystal structure it was concluded that the Arg481 position has been physicochemically conserved on the CSFV helicase with a lysine residue, while the Arg393 position has been replaced with a Ser residue, which is still fully capable of establishing strong hydrogen bonding interactions with potential inhibitor compounds (Figs. 1 and 11).

Figure 6 The seed structure for the drug design experiment.

This is a mercaptoethanol molecule that has been co-crystallized on the HCV helicase crystal structure. Choosing the right starting point for the growing algorithm is very influential to the algorithm. Here the seed was included in the PDB file (1A1V).

The HCV Cys431 residue constitutes an ideal target for drug design experiments not just due to the fact that it had already covalently reacted with a mercaptoethanol molecule in the HCV crystallization solution, but also because its position has been fully conserved on the CSFV helicase too (Fig. 1). More strikingly, the structural superposition of the Dengue helicase with the HCV and CSFV helicases, revealed that the Cys position is conserved on Dengue too, which belongs to the flavivirus genera. Notably, HCV belongs to the hepacivirus genera and CSVF belongs to the pestivirus genera. So, we have three members of the flaviviridae, all from distinct separate generas that still maintain the Cys position conserved. Therefore, upon careful selection of target residues on the helicases of various species within flaviviridae, there is potential for the development of agents that may be active against more than one helicase enzyme.

Figure 7 The dimensions of the ssRNA channel of HCV. Distances and the available space in the Helicase’s active site.

All newly designed compounds should have dimensions that could be accommodated in the above showing area and interact with Arg393 and one or both of the Cys431 and Arg481 residues.

Figure 8 Lead compound from LigBuilder docked into the helicase.

The resulting molecule from the drug design experiment contains 14 chiral centers and is therefore impossible to synthesize. It had to be reduced to a manageable synthetically molecule that still maintains most of the original interactions with the helicase enzyme.

Structurally, the aim was to build a bridge between the Cys431 and the Arg393 (Fig. 7). This compound should covalently interact with the two previous residues and stand in the way of the nucleic acid. In the HCV case, an ideal compound would interact with the two Arg393, Arg481 and the Cys431, thus forming a bridge in the middle of the RNA channel in the helicase. Notably the physical dimensions of the selected site do not change upon the molecular dynamics simulations. If the compound covalently bonds to the receptor, then it is expected to be strong enough to block the passage of the ssRNA thus inhibiting the helicase (Fig. 7). The lead compound designed by LigBuilder expands from Arg393 through to Arg481, via Cys431 (Fig. 7). It fits the available accessible area of the helicase structure and interacts with even more residues than the three target-residues (Fig. 8). Even though LigBuilder suggested it as a lead compound, it is far from being considered as a lead compound, when judging from a drug design point of view. This compound is only an indication of the available interactions that could be potentially established between an inhibitor and the HCV helicase. The process of converting the LigBuilder’s compound into a viable lead from a medicinal chemistry point of view requires manual intervention to substitute bulky groups with those capable of establishing the same type of interaction while being simpler in structure. Figure 8 shows the lead derived from LigBuilder and the interactions it establishes with the helicase. One of the most important properties of an in silico designed lead compound is to be feasible to synthesize. The lead must be synthesizable in order to be eventually tested for its activity. This is essential in order to feedback the drug design process. Even though the lead that LigBuilder generated was capable of establishing so many interactions, it had to be rejected since it would be impossible to synthesize and test such a compound in the lab (Fig. 6). The algorithm needs to understand that the available space is not the whole, extensive oligonucleotide channel of the helicase, but is spatially defined by the three target residues.

That was achieved by physically incorporating a tube-like structure which encloses all key residues and defines the available space for the algorithm to work with (Fig. 9). The tube was originally made from carbon atoms. The PDB file of the tube was edited and the atoms were converted from C (Carbon atom) to Du (Dummy atom). Du atoms do not have any atomic properties and can be recognized by Ligbuilder‘s growing algorithm as inert atoms. They are incapable of establishing interactions of any nature and they have no charge. Their only function is to define the space that is available for the genetic algorithm to work with. The structures that LigBuilder generated proved to be a lot simpler and much more drug-like. The lead that LigBuilder proposed was completely different from the one without the aid of the “tube”. The custom made lead was isolated and then docked again into the original receptor, where it originated from (Fig. 10). Only this time the “tube” was not present and the full receptor was available to be explored by the docking algorithm.

Figure 9 Drug design using the tube spatial restrain.

The application of the “tube” structure that simplified the task of LigBuilder by limiting the available 3D conformational space that new compounds can occupy.

Figure 10 The in silico designed lead compounds.

(A)–(B) The two lead compounds positioned in the ssRNA channel of the HCV helicase. These compounds were designed by using LigBuilder and the “tube”. (C) The mode of interaction between the best, most promising LigBuilder compound and the three target residues of the HCV ssRNA channel.

The result of the docking confirmed that this compound that had been specifically designed for the particular area (between Arg481-Cys431-Arg393) on the helicase found its way to the suggested site and managed to establish the interactions that it was expected to (Fig. 10). The top ranked compound bears an extra SH3 substitution on one of the phenyl rings. That SH3 moiety was able to interact with the Sulfur of the Cys431 and further stabilize the docking. The stability of these compounds derives from their simplicity and rigidity. The conjugated bond and the two phenyl rings on those two compounds do not allow for any flexibility. Figure 10 shows the two versions of the LigBuilder lead compound with and without the CH4 substitution in one of the phenyl rings. It is obvious that the presence of the extra carbon has pushed the compound a bit lower, which is evidence that the extra carbon successfully established an interaction with the nearby available sulphur from the Cys431 residue.

This way the complexity of the structure was forced to be low, by choosing small molecular weights and by limiting the number of interaction between the future compound and the protein to those found by the multi-fragment search. The number of hydrogen bond donors and acceptors was estimated by the Lipinski’s rule of five initially and later on by limiting the number of H bonds to the ones already acquired by the multi-fragment search. This made the drug design approach using the linking algorithm very fast and ensured that all new suggested compounds will be simple in structure and thus synthesizable.

3D Pharmacophore Elucidation and the DNP-poly(A) substrate

3D Pharmacophore designing methods take into account both the three-dimensional structures and binding modes of receptors and inhibitors in order to identify regions that are favorable or not for a specific receptor-inhibitor interaction. The description of the receptor-inhibitor interaction pattern is determined by a correlation between the characteristic properties of the inhibitors and their biochemically determined enzymatic activity.

Based on the findings of the structure based, de novo drug design experiment, a 3D pharmacophore was generated for the active site of CSFV, as a molecular model that ensembles all steric and electronic features that are necessary to ensure optimal covalent and non-covalent interactions with CSFV (Fig. 11). The pharmacophoric features investigated included positively or negatively ionized regions, hydrogen bond donors and acceptors, aromatic regions and hydrophobic areas. For the pharmacophore elucidation process for CSFV’s ssRNA channel all 1200 de novo generated compounds from the drug design experiment were used in their in silico docked conformations. Moreover, a structure based 3D pharmacophore was created using the ssRNA interacting residues on the CSFV helicase channel. The final pharmacophore model elucidated for CSFV helicase was the result of the overlaying of two different pharmacophores that were then reduced to their shared features. This way only their common set of interactions was retained. Our complex-based pharmacophore used a query set that represented a set of receptor-inhibitor interaction fingerprints which were in the form of docked in silico virtual inhibitor complexes.

Figure 11 The pharmacophore model for the HCV template and the CSFV model.

(A) The HCV template with the bound oligonucleotide (white ribbon). (B) The 3D pharmacophore in the presence of the key HCV residues (in ball and stick representation) and the corresponding CSFV amino acids (in wire representation). The pharmacophore color coding is blue for hydrogen acceptor, green for hydrophobic or aromatic rings and magenta for hydrogen donors. (C) The final pharmacophore model for the CSFV helicase.

It was determined that a set of criteria had to be satisfied in order for a candidate inhibitor compound to be active for the CSFV and HCV helicase. First, using the HCV helicase’s 3D structure, there should be one electron donating group (Fig. 11, magenta color) in the proximity of the Arg393 residue (or the corresponding lysine on the CSFV helicase model). Moreover three electron donating groups should be present in the proximity of both Arg393 and Arg481 amino acids (or the corresponding lysine and aspartic acid on the CSFV helicase model). Those interaction sites may not strictly represent hydrogen bonds but water or ion mediated bridges too. Finally the space in between Arg393 and Arg481 in the HCV or the corresponding Lys-Asp residues on the CSFV helicase should be occupied by a large conjugated set of either hydrophobic or aromatic rings. Notably the physical dimensions of this site do not change on either the HCV or CSFV model upon molecular dynamics simulations. Collectively, according to our in silico prediction model, a potent candidate inhibitor of the CSFV helicase should satisfy all of the previously described pharmacophoric features.

Conclusions

The 3D model of the Classical Swine Fever virus helicase was designed using the homologous X-ray crystal structure of the Hepatitis C viral helicase as a template. The model was successfully evaluated in silico in terms of its geometry, fold recognition and compliance to the criteria required as a member of the Flaviviridae virus family. Furthermore, a comprehensive 3D pharmacophore model was constructed alongside the proposition of a lead compound as an in silico predicted potential inhibitor for this enzyme. It is therefore proposed that the Classical Swine Fever virus helicase model will be suitable for further in silico structure-based de novo drug design experiments. Future experiments may lead to the screening of large chemical compound libraries in search of a potent inhibitor. These computer-based methodologies are now becoming an integral part of the drug discovery process that may eventually lead to the development of potential inhibitor structures against the Classical Swine Fever viral helicase.

Supplemental Information

Figure S1 All residues surrounding the ssRNA fragment, color-coded as Figure 2

The ssRNA interacting regions are almost identical between the Classical Swine Fever virus helicase model and the Hepatitis C helicase template.

Click here for additional data file.

Figure S2 The dihedral (omega, planarity and C-beta) and energy plots for the HCV template structure (PDB entry: 1A1V)

(A) The plot is showing the omega torsion profile. (B) The profiles of phi/psi angles. (C) The planarity the third the C-beta torsion angle profile. (D) the rotamer strain energy profile in kcal/mol. The HCV helicase sequence is on the X-axis.

Click here for additional data file.

Figure S3 The dihedral (omega, planarity and C-beta) and energy plots for the CSFV homology model after the molecular dynamics simulations

(A) The plot is showing the omega torsion profile. (B) The profiles of phi/psi angles. (C) The planarity the third the C-beta torsion angle profile. (D) the rotamer strain energy profile in kcal/mol. The CSVF helicase sequence is on the X-axis.

Click here for additional data file.

Additional Information and Declarations

Competing Interests

Author Contributions

The authors declare no conflict of interest.

Dimitrios Vlachakis and Sophia Kossida conceived and designed the experiments, performed the experiments, analyzed the data, contributed reagents/materials/analysis tools, wrote the paper.

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
