# Peer review of "Molecular modeling and pharmacophore elucidation study of the Classical Swine Fever virus helicase as a promising pharmacological target"

_PeerJ, doi:10.7717/peerj.85_

## Round 0.1 · original submission · Major Revisions

As you can see below from reviewers’ comments and feedback, the major concern about this manuscript is the lack of solid validation of new research findings (computational and/or experimental validation), and the the testability of the reported results. The current version of the manuscript is not acceptable for publication unless solid validation has been added, and all the reviewers’ comments have been addressed sufficiently.

Reviewer 1 ·

Basic reporting

In this work, the authors model the structure of the helicase of the Classical Swine Fever Virus (CSFV) based on the template of the Hepatitis C Virus (HCV) helicase and, after checking its quality, they use it for designing a potential inhibitor. Though the techniques used are not novel, the protein studied has practical relevance and the study is well presented.
An important issue concerns how the modeled coordinates andmade available to the community.

Experimental design

The techniques employed are sound, however some points could be improved, or at least discussed.
- When the sequence identity is not high, such in this case, Hidden Markov Models produce much better alignments than programs based on a fixed scoring matrix, such as the one applied by the authors. Did you try any HMM based-program?
- On pages 5 and 6, two different sequence identities are reported between the target sequence and the two sequence identical templates. I deduce from this result that the authors used the sequence of the solved residues for the alignment (the structure of 1a1v has 3 disordered residues), however in my experience a better alignment is obtained aligning the full length sequence.
- The two structures of the HCV helicase found in the PDB present a large conformation change (> 3 angstrom), in this case I would not use both structures with Modeller but rather I would build two separate models of the two conformations.
- The Modeller suite provides the DOPE energy profile to assess the quality of the model on a residue-residue basis (the profile of the model and the template should be compared to identify regions that may be improperly aligned or modeled)

Validity of the findings

The results presented are sound, and the quality of the model seems to be sufficient for the task of the authors to design new drugs.

Additional comments

The paper is well written.
In the caption of Fig.2 the PDB code is wrong (1v1a -> 1a1v)
I suggest to place the section "Electrostatic potential surfaces" before "Structure based drug design", to increase the fluidity of the text
Typos and unclear expressions:
page 15, line 18: was subjected
page 15, line 22: between with
P.18, l.8-9: is requires
P.18, l.16-17: Better write "since it would be impossible to synthesize and test such a compound in the lab"
P.18, l.18: There is a need of giving the algorithm

·

Basic reporting

The manuscript text is presented clearly, however the figures and figure captions are poor. It is hard to see what we are meant to get out of many of the figures and generally not enough detail is given in the
captions:

Fig 2C is not clear, perhaps using a white background rather than
black would mke the foreground stand out more?

Fig 2B, RNA is in blue, not magenta

Fig 3 it is also not clear what we are meant to get out of this. I cannot
see where ATP binds, and the highlighted regions do not sit beside
the ssRNA

Fig 7. A single well chosen panel would be sufficient here

Fig 9. Explain the colour coding of the pharmacophores

Experimental design

None

Validity of the findings

This manuscript describes an interesting attempt to build a homology
model of the Classical Swine Fever Virus helicase protein and to
design potential inhibitors for it. While the individual steps are
sound, each introduces a separate degree of uncertainty, which makes
me a little skeptical as to the relevance of the final
result. However, it may provide a starting point for future work in
the area. I had 2 major concerns which I will describe first, before
noting some more minor points.

1. The justification for choosing the specific inhibitor binding site
needs to be expanded. it is claimed that this site may act to block
the ssRNA binding site, but no more reason than this is given. The
case for choosing this site would be much stronger if the authors
could say something about the site of known inhibitors of CSFV or
HCV. Are there any known inhibitors of these proteins? What is known
about the location of their binding sites? What is the significance of
mercaptoethanol in the HCV structure? Why was it present in the
crystal? Is this a known inhibitor of HCV?

2. The drug molecules are chosen to straddle an 18A long space between
two arginine residues with limited interactions between these sites,
along with binding to a single cys residue. Given that this site is
not a well formed pocket and invovles a lot of empty space (a
requirement given that the ssRNA binds in there), I could not help but
feel that the specificity of the site is bound to be low. I am
skeptical that an 18A long flexible molecules such as the lgands that
are proprosed will bind in this position with enough affinity. The
thing that lends this plausibility is that if the covelent bond to the
Cys residue can be formed then this will limit the avaialble positions
for the compound. Without this I doubt there is much hope the drug
would find its target. So my question is this: What evidience is there
that the ligand can be made to find and covalently bind with the Cys
residue? Any evidence here would strengthen the case for using the
proposed ligands as leads.

Additional comments

Why is the sequence identitiy of the CSFV helicase different to 1A1V and 8OHM? Were the crystalised structures obtained from different organisms?

p6 Clarify the statment: '...only margnially allow for conventional
homology model techniques...' Perhaps it is clearer to say that this is
on the lower boundary at which conventional homology modelling
techniques are applicable.


What does it mean to say the simulations are conducted at 1atm when
you are running in NVT ensemble? In fact there is no specified
pressure under NVT.



p11, It is stated that a method is used to the determine 'most active
compounds'. How is this done? Some more explanation is required as
this seems like an impossible task.


Did the drug docking studies use the post MD model, or the original
homology model?

The electrostatics section appears out of place as it is used to
support the homology model not the drug binding studies. I suggest it
should come prior to drug design section.

How much does the distance between the ARG residues change during the
100ns MD? This may give some idea of the stability of the proposed
binding site.

Reviewer 3 ·

Basic reporting

Adequate. See Below.

Experimental design

There is no experimental validation of the results nor comparative analysis of methods or a substantive integration of existing concepts and/or data or indeed anything than allows this to be classed as science and therefore publishable.

Validity of the findings

There is no experimental validation of the results nor comparative analysis of methods or a substantive integration of existing concepts and/or data or indeed anything than allows this to be classed as science and therefore publishable.

Additional comments

There is no experimental validation of the results nor comparative analysis of methods or a substantive integration of existing concepts and/or data or indeed anything than allows this to be classed as science and therefore publishable. The modelling must reconcile facts or make testable predictions which you go onto test and the results reported. This kind of exercise has no value in itself other than in relation to experimental data.

---

## Round 0.2 · Minor Revisions

Many thanks for resubmitting your revised paper, and thanks for the effort made to improve the paper and address the issues raised by the reviewers’ comments and feedback.

I would like to invite you to address the following three issues (as minor revisions): (1) for the in silico experimental validation of your findings, you provided four figures as supplementary information which is very useful, but I would like you to include some of them into the main text, to provide discussion/justification about the validation. Careful annotations on the figures would be useful too; (2) if possible I would you to provide open-access for the 3D CSFV helicase models, rather than by email request. This will be important in order for enable readers appreciate the value of the work and to extend the work if possible; (3) to carefully address the possible overlapping of information between your recently publication on PeerJ (2013; 1:e74).

---

## Round 0.3 · accepted · Accept

Many thanks for your effort made to enhance the quality of figures and the correction on the figures and manuscript, and for providing open-access for the 3D CSFV helicase models. I am now happy to accept the paper for publication.